# INFOIGL: INVARIANT GRAPH LEARNING DRIVEN BY INFORMATION THEORY

## ABSTRACT

Graph-based tasks often violate the i.i.d. assumption as data collection scenarios change, having attracted significant attention to graph out-of-distribution (OOD) generalization. While extracting invariant features is a popular solution, existing methods are limited by the complexity of graphs with distribution shifts in both attributes and structures. Moreover, identifying invariance on graphs is challenging due to lack of prior knowledge for the invariant features. To address these problems, we propose a novel framework, InfoIGL, which leverages information theory to extract invariant graph representations. The framework treats mutual information as the invariance of graphs by exploiting rich *semantic* relations among different distributions. Specifically, InfoIGL decomposes the process of extracting invariant features for graphs into two tasks: **reducing redundant information** and **maximizing mutual information**. To reduce redundancy, InfoIGL leverages attention mechanism to reduce the entropy of graph representations through optimizing their probability distribution. Then InfoIGL integrates semantic-wise and instance-wise contrastive learning to maximize mutual information through joint optimization. Additionally, instance constraint and hard negative mining are utilized to avoid the collapse of contrastive learning. Experiments on both synthetic and real-world datasets demonstrate that our method achieves state-of-the-art performance under OOD generalization for graph classification tasks. The source code is available at https://anonymous.4open.science/r/InfoIGL-268D.

## 1 INTRODUCTION

Graphs are ubiquitous in the real world, appearing as chemical molecules (Fout et al., 2017), social networks (Wu et al., 2020), and knowledge graph (Hamaguchi et al., 2017), to name a few examples. In recent years, graph neural networks (GNNs) (Kipf & Welling, 2017; Velickovic et al., 2017; Xu et al., 2018) have emerged as a potent representation learning technique for analyzing and making predictions on the graph structures. Despite significant advancements, most existing GNN approaches rely heavily on the i.i.d. assumption that the distribution of test data is independently and identically distributed to the training data. Such an assumption, however, seldom holds in practice due to the spatial-temporal heterogeneity nature of graph, thus having attracted significant attention in addressing the out-of-distribution (OOD) problem.

Identifying graph features that remain invariant across distribution shifts (Koyama & Yamaguchi, 2020; Wang & Veitch, 2022; Wu et al., 2022; Liu et al., 2022b) is paramount in overcoming the graph OOD problem. Recent studies focused mainly on developing methods in the principle of invariance, which can be roughly categorized into two groups. On one hand, *graph manipulation approaches* (Miao et al., 2022; Rong et al., 2019; Wang et al., 2021; Han et al., 2022) typically first generate diverse augmented data (*e.g.,* adding or removing nodes and edges) to increase the coverage of data distribution, then learn representations consistent for all manipulated versions. However, accurately performing data augmentations can be non-trivial, particularly when dealing with complex graphs, of which the failure may destroy the invariant features. In contrast, *causal disentanglement methods* aim to extract the underlying causal subgraphs responsible for generating consistent output across different distributions, utilizing techniques such as stable learning (Fan et al., 2021; Li et al., 2021) and causal intervention (Wu et al., 2022; Sui et al., 2022; Liu et al., 2022a). Nevertheless, it requires prior knowledge regarding causal subgraphs, which may be difficult to obtain in practice. As shown in Figure 1, the performance improvements of representative methods belonging to the two

groups are faint as compared with traditional ERM (Montanari & Saeed, 2022). To circumvent the aforementioned limitations, in this work, we exploit information theory to effectively identify invariant features within complex graph data. Specifically, on the concept of mutual information(Latham & Roudi, 2009), we first quantify the shared information among the samples within the same class, then isolate the common patterns that determine invariance, and extract the invariant features. However, two major challenges need to be addressed:

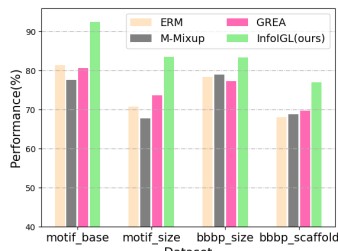

Figure 1: Performance comparison. M-Mixup (Wang et al., 2021) and GREA (Liu et al., 2022a)are representative methods in data manipulation and causal disentanglement approaches respectively.

- Quantifying mutual information for intricate graphs poses great challenges due to high-dimensional continuous variables and unknown data distributions, which might significantly decrease the model efficiency.
- Directly maximizing the mutual information of the extracted features may be misled by redundant information, thereby leading to inaccurate estimation and exacerbating distribution shift issues.

To bridge the gap, we propose a novel framework **InfoIGL** for graph OOD generalization, which decomposes the invariant feature extraction process into two key tasks driven by information theory: **redundant information reduction** to compress the redundancy of the extracted graph features, and **mutual information maximization** to preserve useful invariant features, both tasks complement each other mutually to achieve effective invariant feature extraction from graphs. Specifically, inspired by the data compression theory (Al-Shaykh & Mersereau, 1998; Hosaka & Kabashima, 2006), InfoIGL first reduces redundancy present in the input graph via an attention mechanism to prevent erroneous identification of redundant information as invariant features. Then InfoIGL utilizes semantic-instance (Zhang et al., 2022b; Yue et al., 2021) and instance-instance (Wang et al., 2022) contrastive learning to maximize the mutual information of positive instances supervised by class labels. To prevent the target collapse issue of contrastive loss (Zhang et al., 2022b; Yao et al., 2022), InfoIGL strengthens instance-wise contrastive learning with instance constraint and hard negative mining techniques. We provide thorough theoretical analyses for these modules from the perspective of information theory and demonstrate the effectiveness of InfoIGL with extensive experiments. Our main contributions are summarized as follows.

- To address graph OOD problem, we propose a novel framework called InfoIGL driven by information theory, which views mutual information as invariant features through realizing two tasks.
- We probe the relationship between semantic-wise and instance-wise contrastive learning to maximize mutual information with adequate theoretical evidence.
- We conduct extensive experiments on diverse benchmark datasets to demonstrate the effectiveness of our proposed framework InfoIGL.

## 2 PROBLEM FORMULATION AND PRELIMINARY

### 2.1 PROBLEM FORMULATION

Let $\mathbb{G}$ and $\mathbb{Y}$ be the sample space and label space, respectively. We denote a graph by $G_i \in \mathbb{G}$ whose adjacent matrix and node feature matrix are $\mathbf{A}$ and $\mathbf{X}$, respectively. A graph predictor $f = \theta \circ \Phi : \mathbb{G} \to \mathbb{Y}$ maps the input graph $G_i$ to the corresponding label $\mathbf{y}_i \in \mathbb{Y}$, which can be decomposed into a graph encoder $\Phi$ and a classifier $\theta$. Let $\mathcal{D}^{tr} = \{(G_i, \mathbf{y}_i)\}_{i=1}^{N^{tr}}$ and $\mathcal{D}^{te} = \{(G_i, \mathbf{y}_i)\}_{i=1}^{N^{te}}$ be the training and testing dataset under distribution shift, *i.e.*, $P^{tr}(G_i, \mathbf{y}_i) \neq P^{te}(G_i, \mathbf{y}_i)$, where $\mathcal{D}^{te}$ remains unobserved during the training stage. Consequently, the generalization for graph OOD can be formulated as follows:

$$\min_{\Phi:G \to H, \theta:H \to Y} \mathbb{E}_{G, \mathbf{y} \in \mathcal{D}^{te}}[\mathcal{L}(\mathbf{y}, \theta \circ \Phi(G)]$$
(1)

To overcome the distribution shifts between $\mathcal{D}^{te}$ and $\mathcal{D}^{tr}$, invariant representation learning aims to optimize the encoder $\Phi(\cdot)$ to capture representations that keep invariant across all distributions when the classifier $\theta$ is deterministic and well performed, *i.e.*, $P^{tr}(Y|\Phi(G)) = P^{te}(Y|\Phi(G))$. Towards this end, we scrutinize the task of invariant feature learning through the lens of information theory. Our fundamental insight is that **the entropy of invariant features is equivalent to the mutual information across graphs of the same class**. By maximizing the mutual information between extracted graph representations, we can approach the ideal mutual information and ultimately obtain representations of invariant features. However, the quantization of mutual information for the extracted graph representations is vulnerable to interference from redundant information present in the graphs. To overcome this challenge, we propose to decompose this task into two subtasks (*cf.* Figure 2): the compression of redundant information, followed by the maximization of mutual information.

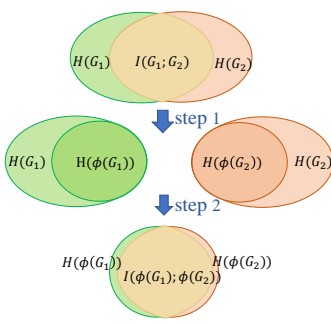

Figure 2: Task decomposition. Step1: Compressing $H(\Phi(G))$ to reduce redundancy; Step2: Maximizing $I(\Phi(G_1); \Phi(G_2))$ to its upper bound $I(G_1, G_2)$

## 2.2 TASK I: REDUNDANT INFORMATION REDUCTION

In this work, we use the term "*redundant information*" to refer to the irrelevant yet potentially confounding features present in graph data. According to Proposition 1, directly quantifying the mutual information without first isolating these extraneous factors may be misled by redundant information (Chuang et al., 2022), resulting in inaccurate estimation.

**Proposition 1.** *Given variables $X_1, X_2$ and nosie $S_1, S_2$, $X$ is independent of $S$, we have: $I(X_1, S_1; X_2, S_2) - I(X_1; X_2) = I(S_1; S_2) \geq 0$. Directly calculating the mutual information between variables with noise may lead to capturing spurious correlations (Chuang et al., 2022).*

To solve this problem, we take inspiration from the concept of "redundancy" in information theory (*cf.* Theorem 1), which guides us to reduce redundant information via data compression.

**Theorem 1.** *In information theory (Shannon, 1948), the redundancy $\gamma$ of variable $X$ is defined as follows: $\gamma = 1 - \eta = 1 - \frac{H_\infty(X)}{H_m(X)}$, where $H_\infty(X)$ is the limiting entropy, $H_m(X)$ is the m-order Markov average entropy.*

In the context of information theory, $H_\infty(X)$ and $H_m(X)$ refer to the average entropy per symbol (*i.e.*, $X$) in a sequence of infinite or $m$ symbols respectively. To extend the concept of redundancy to graph learning, we posit that the entropy of graph representations generated by an ideal graph encoder through infinite training data (*i.e.*, $H(\Phi_\infty(G))$) can be viewed as the limiting entropy, while its counterpart obtained from the training set (*i.e.*, $H(\Phi(G))$) can be viewed as $m$-order Markov average entropy. Thereby, the goal of task I is to reduce $H(\Phi(G))$ close to $H(\Phi^*(G))$.

## 2.3 TASK II: MUTUAL INFORMATION MAXIMIZATION

Mutual information assesses the level of shared information between variables, which can serve as a measure of invariance across different graphs. Ideally, if the mutual information between graph representations generated by graph encoder $\Phi$ reaches its upper bound $I(G_1; G_2)$, it can be considered as the optimal graph encoder $\Phi^*$ for extracting invariant features (Shwartz-Ziv & Tishby, 2017; Tian et al., 2020), that is, $\Phi^* = \underset{\Phi}{\arg\max} I(\Phi(G_1); \Phi(G_2))$. However, directly maximizing mutual information poses significant challenges due to the high dimensionality of continuous variables and the unknown nature of data distributions. To overcome these obstacles, it is possible to establish tractable objectives that approaches the boundary of mutual information.

**Theorem 2.** *(Mutual information boundary (Oord et al., 2018; Poole et al., 2019)) Given two variables $X, Y$, the bound of mutual information between $X$ and $Y$ can be defined as follows:*

$$I(X;Y) \geq E[\frac{1}{K} \sum_{i=1}^{K} \log \frac{\exp(f(x_i, y_i)))}{\frac{1}{K}\sum_{j=1}^{K} \exp(f(x_i, y_j))}] \triangleq I_{NCE} \quad (2)$$

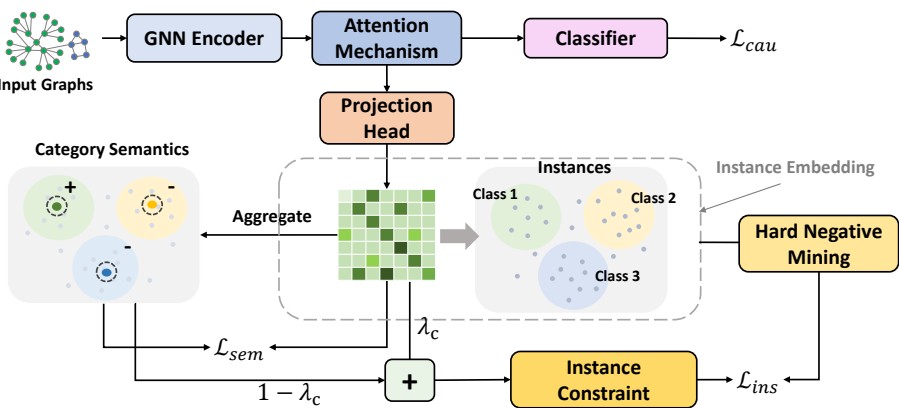

Figure 3: The proposed InfoIGL framework.

*where $x_i$ and $y_i$ denote a positive pair sampled from the joint distribution $P(X, Y)$, while $x_i$ and $y_j$ form a negative pair sampled from the product of marginal distributions $P(X)P(Y)$.*

As defined in Equation equation 2, maximizing the mutual information equals to minimizing the InfoNCE loss (Oord et al., 2018), which inspires us to leverage contrastive learning to accomplish Task II.

## 3 METHOD AND THEORETICAL ANALYSIS

In this section, we introduce our framework InfoIGL that simultaneously addresses two tasks to extract invariant graph representations, as illustrated in Figure 3. Specifically, (1) For task I, we utilize attention mechanism to reduce redundancy. (2) For task II, we jointly optimize the semantic-instance and instance-instance contrastive learning to maximize the mutual information. (3) Finally, we transfer the invariant representation to the downstream task — graph classification. Task I and Task II are complementary to each other. Reducing redundancy can prevent capturing spurious correlations, while maximizing mutual information can avoid discarding useful information.

### 3.1 TOWARDS REDUNDANT INFORMATION REDUCTION

To reduce redundancy, we hire attention mechanism (Sui et al., 2022; Brody et al., 2022; Thekumparampil et al., 2018; Kim & Oh, 2022) to assign weights to nodes and edges according to the Theorem 3.

**Theorem 3.** *The entropy of variable $X$ (i.e., $H(X) = -\sum p(x_i)log_2 p(x_i)$) is a convex function, which can be reduced through optimization of the probability distribution $P(X)$.*

In the context of graphs, we can reduce the entropy of graph representations by optimizing the probability distribution of features, which can be realized through attention mechanism (Sui et al., 2022; Brody et al., 2022). First, we obtain the representations for graph nodes with GNNs. Taking GIN (Xu et al., 2018) as an example, the node update module is defined as follows:

$$\mathbf{h}_v^{(k)} = \text{MLP}^k((1 + \epsilon^{(k)}) \cdot \mathbf{h}_v^{(k-1)} + \sum_{u \in N(v)} \mathbf{h}_u^{(k-1)}) \tag{3}$$

where $\text{MLP}(\cdot)$ stands for multi-layer perceptron, $\epsilon$ is a learnable parameter, $\mathbf{h}_v$ and $\mathbf{h}_u$ separately denote the representations of nodes $v$ and $u$, $N(v)$ denotes the neighbour nodes of $v$.

The attention score of node $v$ and edge $(u, v)$ can be obtained as follows:

$$\alpha_v = \text{softmax}(\frac{Q_v K_v^\top}{\sqrt{d_k}})V_v, \quad \alpha_{uv} = \text{softmax}(\text{LeakyReLU}(\text{MLP}(\mathbf{h}_u || \mathbf{h}_v))) \tag{4}$$

$Q_v = \mathbf{h}_v \cdot \mathbf{W}^Q, K_v = \mathbf{h}_v \cdot \mathbf{W}^K, V_v = \mathbf{h}_v \cdot \mathbf{W}^V, \mathbf{W}^Q, \mathbf{W}^K, \mathbf{W}^V$ are trainable parameter matrices, $d_k$ is $K_v$'s dimension, $\|$ is the concatenation operation.

After assigning weights for nodes and edges, we encode graphs into graph-level representations $\mathbf{h}_G$:

$$\mathbf{h}'_v = \alpha_v \cdot \mathbf{h}_v, \mathbf{h}'_{vu} = \alpha_{uv} \cdot \mathbf{h}_{uv}, \quad \mathbf{h}_G = \text{READOUT}(\mathbf{H}_v, \mathbf{H}_{uv}) \tag{5}$$

where $\mathbf{h}_v, \mathbf{h}_{uv}$ are node and edge embedding obtained from GIN, $\mathbf{H}_v = [..., \mathbf{h}'_v, ...]_{v \in \mathcal{V}}^\top, \mathbf{H}_{uv} = [..., \mathbf{h}'_{uv}, ...]_{u,v \in \mathcal{V}}^\top$, $\mathcal{V}$ denotes the node set. Till now, we have optimized the probability distribution of graph features and reduced the entropy of graph representations.

## 3.2 Towards Mutual Information Maximization

According to Theorem 2, we can leverage contrastive learning to maximize the mutual information between representations. However, traditional contrastive-based methods may not effectively work since complex instance-instance relationships may impede model generalization(Yao et al., 2022; Xu et al., 2021; Zhang et al., 2022c). To fully unleashing the advantages of contrastive learning, we jointly optimize semantic-wise and instance-wise contrastive losses after projecting the embedding to another space with projection head, which can promote both inter-class separation and intra-class compactness.

### 3.2.1 Projection Head

A projection head is a small network that maps the embedding to another space where further contrastive learning is applied (Yao et al., 2022; Gupta et al., 2022), which can prevent the conflicts between reducing representation entropy and strengthening contrastive uniformity according to the Proposition 2.

**Proposition 2.** *Contrastive learning can promote embedding uniformity (Wang & Isola, 2020), which may increase entropy (Ma et al., 2023) and violate task I. Using a projection head as a springboard can avoid potential entropy increase in the original embedding $\mathbf{h}_G$.*

We employ a two-layer MLP $f_{\theta_p}(\cdot)$ as the projection head for $\mathbf{h}_G$:

$$\mathbf{z}_G = f_{\theta_p}(\mathbf{h}_G). \tag{6}$$

Then InfoIGL applies contrastive learning and encourages uniformity in new space $\mathbf{z}_G$.

### 3.2.2 Semantic-wise Contrastive Learning

Semantic-instance contrastive learning is more robust to complex instances by neutralizing the noise in multiple instances. To realize semantic-wise contrastive learning, we first introduce the cluster center of each class as the corresponding category semantic. Formally, we initialize the semantic $\mathbf{w}_c$ as the average semantic representation over examples belonging to class c: $\mathbf{w}_c = \frac{1}{N_c} \sum_{i=1}^{N_c}(\mathbf{z}_{G_i})$, $N_c$ is the number of $\mathbf{z}_G$ with label c in a batch. Then we update the current round $\mathbf{w}_c^{(r)}$ by calculating the similarity between each instance embedding $\mathbf{z}_{G_i}$ and the semantic of last round $\mathbf{w}_c^{(r-1)}$:

$$m_i^{(r)} = \text{softmax}(\text{cosine}(\mathbf{z}_{G_i}, \mathbf{w}_c^{(r-1)})), \quad \mathbf{w}_c^{(r)} = \sum_{i=1}^{N_c} m_i^{(r)} \cdot \mathbf{z}_{G_i} \tag{7}$$

where $\text{cosine}(\cdot)$ denotes the cosine similarity. Then we define the semantic-instance contrastive loss:

$$\mathcal{L}_{\text{sem}} = -\frac{1}{N^{tr}} \sum_{i=1}^{N^{tr}} \log \frac{\exp(\mathbf{z}_{G_i}^\top \mathbf{w}_c / \tau)}{\exp(\mathbf{z}_{G_i}^\top \mathbf{w}_c / \tau) + \sum_{k=1, k \neq c}^{C-1} \exp(\mathbf{z}_{G_i}^\top \mathbf{w}_k / \tau)} \tag{8}$$

where $N^{tr}$ denotes the number of graphs in a batch, $\mathbf{w}_c$ denotes the target category semantic of $\mathbf{z}_{G_i}$, $C$ denotes the number of classes, $\tau$ is the scale factor.

### 3.2.3 Instance-wise Contrastive Learning

However, merely leveraging semantic-wise contastive learning may sacrifice some potential useful semantic relations (Yao et al., 2022) according to the Proposition 3.

**Proposition 3.** *The mutual information between semantic $\mathbf{w}_c$ and instance $\mathbf{z}_G$ is lower than the mutual information between instance $\mathbf{z}_G$: $I(\sum_{i=1}^{N_c}(m_i \cdot \mathbf{z}_{G_i}); \mathbf{z}_{G_1};..;\mathbf{z}_{G_{N_c}}) \leq I(\mathbf{z}_{G_1}; ...; \mathbf{z}_{G_i}; ...; \mathbf{z}_{G_{N_c}}).$*

To compensate for the missing mutual information, we introduce instance-instance contrastive learning. The loss function for instance-instance contrastive learning can be defined as follows:

$$\mathcal{L}_{\text{ins}} = -\frac{1}{N^{tr}} \sum_{i=1}^{N^{tr}} \log \frac{\exp(\mathbf{z}_{G_i}^{\top} \mathbf{z}_{G_+}/\tau')}{\exp(\mathbf{z}_{G_i}^{\top} \mathbf{z}_{G_+}/\tau) + \sum_{k=1}^{K} \exp(\mathbf{z}_{G_i}^{\top} \mathbf{z}_{G_k}/\tau')} \tag{9}$$

Positive sample $\mathbf{z}_{G_+}$ are randomly selected from graphs that belong to the same class as $\mathbf{z}_{G_i}$, $K$ denotes the number of negative samples for each graph instance.

### 3.2.4 DISCUSSION FOR INSTANCE-INSTANCE CONTRASTIVE LEARNING

For instance-instance contrastive learning, model collapse (Jing et al., 2021)(*i.e.,* samples are mapped to the same point) is not uncommon due to excessive alignment. So we apply instance constraint (Yao et al.) and hard negative mining (Xuan et al., 2020; Robinson et al., 2020) to prevent getting stuck in trivial solutions (Yao et al., 2022).

**Instance constraint.** Enhancing the uniformity (Wang & Isola, 2020) of graph embeddings $\mathbf{z}_G$ can prevent model collapse from excessive alignment, which can be assisted by the uniformity of semantics $\mathbf{w}_c$. The semantics $\mathbf{w}_c$ are ensured to be uniformly distributed by semantic-wise contrastive learning. Thus we utilize instance constraint:

$$\mathbf{z}'_G = \lambda_c \cdot \mathbf{z}_G + (1 - \lambda_c) \cdot \mathbf{w}_c \tag{10}$$

where $\mathbf{w}_c$ refers to the corresponding semantic belonging to the same class as $\mathbf{z}_G$.

**Hard negative mining.** Hard negative pair (Xuan et al., 2020; Robinson et al., 2020) plays an important role in contrastive loss, which can help the network learn a better decision boundary. We identify hard negative graph samples according to the Definition 1.

**Definition 1.** *For sample $\mathbf{z}'_{G_i}$, there are two principles to mine hard negative samples: 1) $y(\mathbf{z}'_{hard}) \neq y(\mathbf{z}'_{G_i})$, y means the graph label; 2). $||\Phi(\mathbf{z}'_{hard}) - \Phi(\mathbf{z}'_{G_i})||^2 \leq \delta$, $\delta$ is a very small value.*

To identify hard negative samples for instances $\mathbf{z}'_G$ belonging to class $c$, we calculate the distance between semantic $\mathbf{w}_c$ and samples from other classes within a batch, then we choose the $K$ nearest ones as hard negative samples $\{\mathbf{z}'_{hard_k}\}_{k=1}^{K}$. Then the loss for instance-wise contrastive learning can be modified as follows:

$$\mathcal{L}_{\text{ins}} = -\frac{1}{N^{tr}} \sum_{i=1}^{N^{tr}} \log \frac{\exp(\mathbf{z}_{G_i}'^{\top} \mathbf{z}'_{G_+}/\tau')}{\exp(\mathbf{z}_{G_i}'^{\top} \mathbf{z}'_{G_+}/\tau') + \sum_{k=1}^{K} \exp(\mathbf{z}_{G_i}'^{\top} \mathbf{z}'_{hard_k}/\tau')} \tag{11}$$

### 3.3 DOWNSTREAM TASK TRANSFER AND OVERALL FRAMEWORK

To make the invariant graph representation applicable for graph classification, we define the loss function for prediction as follows:

$$\mathcal{L}_{\text{pred}} = -\frac{1}{N^{tr}} \sum_{i=1}^{N^{tr}} \mathbf{y}_i^{\top} \log(\theta(\mathbf{h}_{G_i})) \tag{12}$$

where $\mathbf{y}_i$ is the label of $G_i$, $\theta$ is the classifier for $\mathbf{h}_{G_i}$.

The overall framework is illustrated in Figure 3 and the final loss can be given by:

$$\mathcal{L} = \mathcal{L}_{\text{pred}} + \lambda_s \mathcal{L}_{\text{sem}} + \lambda_i \mathcal{L}_{\text{ins}} \tag{13}$$

where the hyperparameters $\lambda_s, \lambda_i$ are scaling weights for each loss, which can adjust the impact of different modules on the model's results. The algorithm of training stage is listed in the Appendix.

# 4 EXPERIMENTS

In this section, we conduct extensive experiments on multiple datasets to answer the following questions: Q1: How effective is InfoIGL compared to prior methods for OOD generalization in graphs? Q2: How do the two tasks make sense respectively? Q3: How do the different contrastive modes impact InfoIGL's performance respectively? Q4: How sensitive the model is to the hyperparameters?

## 4.1 DATASETS AND BASELINES

We conduct experiments on one synthetic (*i.e.,* Motif) and three real-world (*i.e.,* HIV, Molbbbp, and CMNIST) datasets which are designed for graph OOD (Gui et al., 2022; Hu et al., 2020), where Motif and CMNIST are evaluated with ROC-AUC while HIV and Molbbbp are evaluated with ACC. We compare our InfoIGL against diverse graph OOD generalization baselines: **Optimization methods:** ERM, IRM (Arjovsky et al., 2019), VREx (Krueger et al., 2021), GroupDRO (Sagawa et al., 2020), FLAG (Kong et al., 2022); **Causal learning:** DIR (Wu et al., 2022), CAL (Sui et al., 2022), GREA (Liu et al., 2022a), CIGA (Chen et al.), Disc (Fan et al., 2022); **Stable learning:** StableGNN (Fan et al., 2021), OOD-GNN (Li et al., 2021); **Data manipulation:** GSAT (Miao et al., 2022), DropEdge (Rong et al., 2019), M-Mixup (Wang et al., 2021), G-Mixup (Han et al., 2022). Since the practical implementation of InfoIGL involves utilizing graph contrastive learning to maximize mutual information, we also incorporate **classical graph contrastive learning** methods as benchmarks, including CNC (Zhang et al., 2022a), GMI (Peng et al., 2020), Infograph (Sun et al., 2019), GraphCL (Hafidi et al.). Detailed explanations for datasets and baselines are provided in the Appendix.

## 4.2 OVERALL RESULTS(Q1)

Table 1: Performance of different methods on synthetic (Motif) and real-world (HIV, Molbbbp, CMNIST) datasets. The best results are in **bold**, and the runner-up results are underlined.

| methods | Motif | | HIV | | Molbbbp | | CMNIST |
|---|---|---|---|---|---|---|---|
| | size | base | size | scaffold | size | scaffold | color |
| ERM | $70.75_{\pm0.56}$ | $81.44_{\pm0.45}$ | $63.26_{\pm2.47}$ | $72.33_{\pm1.04}$ | $78.29_{\pm3.76}$ | $68.10_{\pm1.68}$ | $28.60_{\pm1.87}$ |
| IRM | $69.77_{\pm0.88}$ | $80.71_{\pm0.46}$ | $59.90_{\pm3.15}$ | $72.59_{\pm0.45}$ | $77.56_{\pm2.48}$ | $67.22_{\pm1.15}$ | $27.83_{\pm2.13}$ |
| GroupDRO | $69.98_{\pm0.86}$ | $81.43_{\pm0.70}$ | $61.37_{\pm2.79}$ | $\mathbf{73.64_{\pm0.86}}$ | $79.27_{\pm2.43}$ | $66.47_{\pm2.39}$ | $29.07_{\pm3.14}$ |
| VREx | $70.24_{\pm0.72}$ | $81.56_{\pm0.35}$ | $60.23_{\pm1.70}$ | $72.60_{\pm0.82}$ | $78.76_{\pm2.37}$ | $68.74_{\pm1.03}$ | $28.48_{\pm2.87}$ |
| FLAG | $56.26_{\pm3.98}$ | $72.29_{\pm1.31}$ | $66.44_{\pm2.32}$ | $70.45_{\pm1.55}$ | $79.26_{\pm2.26}$ | $67.69_{\pm2.36}$ | $32.30_{\pm2.69}$ |
| DIR | $54.96_{\pm9.32}$ | $82.96_{\pm4.47}$ | $72.61_{\pm2.03}$ | $69.05_{\pm0.92}$ | $76.40_{\pm4.43}$ | $66.86_{\pm2.25}$ | $33.20_{\pm6.17}$ |
| CAL | $66.64_{\pm2.74}$ | $81.94_{\pm1.20}$ | $\underline{83.33_{\pm2.84}}$ | $73.05_{\pm1.86}$ | $79.20_{\pm3.81}$ | $67.37_{\pm3.61}$ | $27.99_{\pm3.24}$ |
| GREA | $\underline{73.31_{\pm1.85}}$ | $80.60_{\pm2.49}$ | $66.48_{\pm4.13}$ | $70.96_{\pm3.16}$ | $77.34_{\pm3.52}$ | $\underline{69.72_{\pm1.66}}$ | $29.02_{\pm3.26}$ |
| CIGA | $70.65_{\pm4.81}$ | $75.01_{\pm3.56}$ | $65.98_{\pm3.31}$ | $64.92_{\pm2.09}$ | $76.08_{\pm1.21}$ | $66.43_{\pm1.99}$ | $23.36_{\pm9.32}$ |
| DisC | $53.34_{\pm13.71}$ | $76.70_{\pm0.47}$ | $56.59_{\pm10.09}$ | $67.12_{\pm2.11}$ | $75.68_{\pm3.16}$ | $60.72_{\pm0.89}$ | $24.99_{\pm1.78}$ |
| GSAT | $64.16_{\pm3.35}$ | $83.71_{\pm2.30}$ | $65.63_{\pm0.88}$ | $68.88_{\pm1.96}$ | $75.63_{\pm3.83}$ | $66.78_{\pm1.45}$ | $28.17_{\pm1.26}$ |
| DropEdge | $55.27_{\pm5.93}$ | $70.84_{\pm6.81}$ | $54.92_{\pm1.73}$ | $66.78_{\pm2.68}$ | $78.32_{\pm3.44}$ | $66.49_{\pm1.55}$ | $22.65_{\pm2.90}$ |
| M-Mixup | $67.81_{\pm1.13}$ | $77.63_{\pm0.57}$ | $64.87_{\pm1.77}$ | $72.03_{\pm0.53}$ | $78.92_{\pm2.43}$ | $68.75_{\pm1.03}$ | $26.47_{\pm3.45}$ |
| G-Mixup | $59.92_{\pm2.10}$ | $74.66_{\pm1.89}$ | $70.53_{\pm2.02}$ | $71.69_{\pm1.74}$ | $78.55_{\pm4.16}$ | $67.44_{\pm1.62}$ | $31.85_{\pm5.82}$ |
| OOD-GNN | $68.62_{\pm2.98}$ | $74.62_{\pm2.66}$ | $57.49_{\pm1.08}$ | $70.45_{\pm2.02}$ | $79.48_{\pm4.19}$ | $66.72_{\pm1.23}$ | $26.49_{\pm2.94}$ |
| StableGNN | $59.83_{\pm3.40}$ | $73.04_{\pm2.78}$ | $58.33_{\pm4.69}$ | $68.23_{\pm2.44}$ | $77.47_{\pm4.69}$ | $66.74_{\pm1.30}$ | $28.38_{\pm3.49}$ |
| CNC | $66.52_{\pm3.12}$ | $82.51_{\pm1.26}$ | $70.68_{\pm2.15}$ | $66.53_{\pm2.19}$ | $76.19_{\pm3.52}$ | $68.16_{\pm1.25}$ | $32.41_{\pm1.28}$ |
| GMI | $67.90_{\pm1.46}$ | $79.52_{\pm0.45}$ | $74.34_{\pm0.55}$ | $\underline{73.44_{\pm0.35}}$ | $77.67_{\pm0.30}$ | $69.38_{\pm1.02}$ | $30.24_{\pm5.98}$ |
| InfoGraph | $67.49_{\pm2.54}$ | $75.57_{\pm0.88}$ | $74.63_{\pm0.80}$ | $71.41_{\pm0.82}$ | $\underline{80.82_{\pm0.49}}$ | $70.39_{\pm1.34}$ | $\underline{33.84_{\pm1.52}}$ |
| GraphCL | $66.90_{\pm2.80}$ | $74.40_{\pm0.90}$ | $77.13_{\pm0.17}$ | $72.94_{\pm0.68}$ | $80.64_{\pm0.78}$ | $69.36_{\pm1.32}$ | $32.81_{\pm1.71}$ |
| InfoIGL(ours) | $\mathbf{85.53_{\pm2.37}}$ | $\mathbf{92.51_{\pm0.16}}$ | $\mathbf{93.15_{\pm0.77}}$ | $72.37_{\pm1.63}$ | $\mathbf{83.39_{\pm2.76}}$ | $\mathbf{77.05_{\pm2.24}}$ | $\mathbf{38.93_{\pm1.11}}$ |
| improvemrnt | ↑ 12.22% | ↑ 8.80% | ↑ 9.82% | ↓ 1.07% | ↑ 2.57% | ↑ 7.33% | ↑ 5.09% |

We train and evaluate our proposed InfoIGL, together with all the baselines, 10 times to obtain the average performance (mean ± standard deviation). Details of hyperparameters are listed in the

Appendix. It can be observed from Table 1 that optimization methods exhibit stable performance with moderate accuracy and low variance, while causal learning baselines show unstable performance with undulating accuracy and high variance. Besides, stable learning and data manipulation baselines perform relatively poorly compared to other baselines. Additionally, traditional graph contrastive learning methods can partially combat distributional shifts, but their effectiveness is not as strong as InfoIGL since they were not designed specially to extract invariant features. These observations indicate that almost all of the baselines have their limitations for graph OOD generalization. Our proposed framework, InfoIGL, achieves state-of-the-art performance on diverse datasets with low variance, outperforming the strongest baseline by 9.82% on HIV (size) and 12.22% on Motif (size). These results demonstrate the effectiveness of InfoIGL in extracting stable and invariant graph representations for graph classification tasks.

### 4.3 ABLATION STUDY FOR Q2

Table 2: Results of ablation experiments on two tasks.

| methods | Motif | | HIV | | Molbbbp | | CMNIST |
|---|---|---|---|---|---|---|---|
| | size | base | size | scaffold | size | scaffold | color |
| ERM | $70.75_{\pm0.56}$ | $81.44_{\pm0.45}$ | $63.26_{\pm2.47}$ | $72.33_{\pm1.04}$ | $78.29_{\pm3.76}$ | $68.10_{\pm1.68}$ | $28.60_{\pm1.87}$ |
| InfoIGL-A | $69.69_{\pm6.24}\downarrow$ | $87.14_{\pm0.88}\uparrow$ | $76.99_{\pm2.55}\uparrow$ | $71.56_{\pm1.96}\downarrow$ | $79.72_{\pm3.50}\uparrow$ | $74.48_{\pm1.00}\uparrow$ | $34.54_{\pm2.11}\uparrow$ |
| InfoIGL-C | $68.01_{\pm2.09}\downarrow$ | $86.63_{\pm1.33}\uparrow$ | $72.81_{\pm2.92}\uparrow$ | $68.02_{\pm2.28}\downarrow$ | $75.32_{\pm1.38}\downarrow$ | $65.62_{\pm1.07}\downarrow$ | $33.40_{\pm2.10}\uparrow$ |
| InfoIGL | $85.53_{\pm2.37}\uparrow$ | $92.51_{\pm0.16}\uparrow$ | $93.15_{\pm0.77}\uparrow$ | $72.37_{\pm1.63}\downarrow$ | $83.39_{\pm2.76}\uparrow$ | $77.05_{\pm2.24}\uparrow$ | $38.93_{\pm1.11}\uparrow$ |

To validate the significance of each task individually, we conduct separate ablation studies on the attention mechanism and contrastive learning. Specifically, we compare InfoIGL with two variations: (1) InfoIGL-A: which includes only the attention mechanism in Task I, and (2) InfoIGL-C, which focuses solely on the contrastive learning in Task II. The results are reported in Table 2. InfoIGL-A outperforms ERM but falls short of InfoIGL, underscoring the significance of contrastive learning and the necessity of Task II. In contrast, InfoIGL-C yields poorer results than ERM and InfoIGL, shedding light on the impact of Task I.

### 4.4 ABLATION STUDY FOR Q3

Table 3: Results of ablation experiments on semantic-wise and instance-wise contrastive learning.

| methods | Motif | | HIV | | Molbbbp | | CMNIST |
|---|---|---|---|---|---|---|---|
| | size | base | size | scaffold | size | scaffold | color |
| InfoIGL-N | $69.69_{\pm6.24}$ | $87.14_{\pm0.88}$ | $76.99_{\pm2.55}$ | $71.56_{\pm1.96}$ | $79.72_{\pm3.50}$ | $74.48_{\pm1.00}$ | $34.54_{\pm2.11}$ |
| InfoIGL-S | $84.77_{\pm2.10}\uparrow$ | $89.93_{\pm0.93}\uparrow$ | $87.30_{\pm1.21}\uparrow$ | $72.12_{\pm1.87}\uparrow$ | $81.97_{\pm1.79}\uparrow$ | $76.76_{\pm3.66}\uparrow$ | $37.31_{\pm1.50}\uparrow$ |
| InfoIGL-I | $80.05_{\pm2.99}\uparrow$ | $90.36_{\pm1.54}\uparrow$ | $91.38_{\pm2.38}\uparrow$ | $63.70_{\pm5.45}\downarrow$ | $74.91_{\pm2.07}\downarrow$ | $70.11_{\pm2.02}\downarrow$ | $35.67_{\pm1.19}\uparrow$ |
| InfoIGL | $\mathbf{85.53}_{\pm2.37}\uparrow$ | $\mathbf{92.51}_{\pm0.16}\uparrow$ | $\mathbf{93.15}_{\pm0.77}\uparrow$ | $\mathbf{72.37}_{\pm1.63}\uparrow$ | $\mathbf{83.39}_{\pm2.76}\uparrow$ | $\mathbf{77.05}_{\pm2.24}\uparrow$ | $\mathbf{38.93}_{\pm1.11}\uparrow$ |

We perform an ablation study to analyze the impacts of semantic-wise and instance-wise contrastive learning respectively. Specifically, we compare InfoIGL with three variations: (1) InfoIGL-N, which does not use contrastive learning; (2) InfoIGL-S, which employs semantic-wise contrastive learning only; and (3) InfoIGL-I, which applies instance-wise contrastive learning only. We report the results in Table 3. Our observations are as follows: 1) Merely applying instance-instance contrastive learning may cause performance degradation, which confirms the Proposition 1 that directly maximizing mutual information between instances may have a negative impact from an experimental perspective. 2) Applying semantic-instance contrastive learning can achieve improvement coherently on diverse

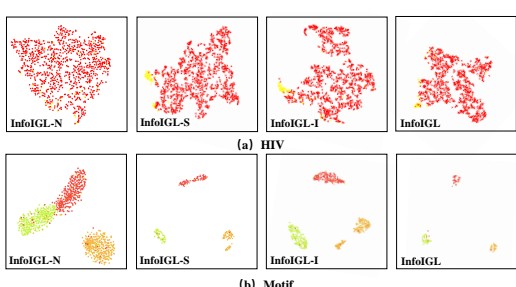

Figure 4: The t-SNE (Van der Maaten & Hinton, 2008) visualizations.

datasets, which demonstrates the robustness of semantic-wise contrastive learning. 3) InfoIGL with both semantic-wise and instance-wise contrastive learning outperforms all of the three variations across diverse datasets, proving that jointly optimizing the two contrastive losses can inspire their individual potentials. Semantic-instance and instance-instance contrastive learning can promote each other and cooperate complementarily.

Additionally, we employ the t-SNE (Van der Maaten & Hinton, 2008) technique to visualize the embedding of graph instances on the HIV (Gui et al., 2022) and Motif (Gui et al., 2022) datasets (*cf.* Figure 4), where four variations (InfoIGL-N, InfoIGL-S, InfoIGL-I, InfoIGL) are compared. The results reveal that compared to InfoIGL-N, the embeddings obtained by InfoIGL-S and InfoIGL-I exhibit a more compact clustering pattern, reflecting the efficacy of semantic-wise and instance-wise contrastive learning in aligning shared information. Furthermore, it is evident that InfoIGL exhibits the best convergence effect, as its embeddings are more tightly clustered than those produced by the other variations.

## 4.5 SENSITIVE ANALYSIS(Q3)

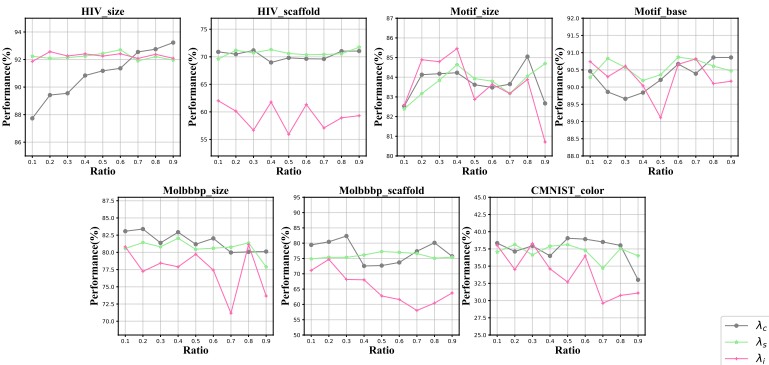

Figure 5: Sensitivity analysis of hyperparameters $\lambda_c, \lambda_s, \lambda_i$

To assess the sensitivity of InfoIGL to its hyperparameters, namely $\lambda_c$ for instance constraint and $\lambda_s$ and $\lambda_i$ for contrastive loss respectively, we conduct sensitivity analysis experiments by tuning these hyperparameters within the range of $\{0.1, 0.2, 0.3, 0.4, 0.5, 0.6, 0.7, 0.8, 0.9\}$ under the controlled experimental setting. Specifically, when adjusting a specific hyperparameter, we fix the remaining hyperparameters at the values that yield the best performance. The results are presented in Figure 5. It is noteworthy that InfoIGL is relatively insensitive to $\lambda_c$ and $\lambda_s$ (gray and green), as adjusting their values does not cause significant fluctuations in InfoIGL's performance. While $\lambda_i$ (pink) has a significant impact on the InfoIGL and requires fine-tuning. Furthermore, the results demonstrate the negative impact of excessively large values for $\lambda_i$ in InfoIGL. For instance, the performance of InfoIGL on Motif (size) and CMNIST (color) drops significantly when $\lambda_i$ exceeds 0.6. By comparing the performance of InfoIGL across different datasets under different hyperparameter settings, we can identify the optimal hyperparameters for each dataset. For example, $\lambda_s$ ranging from 0.3 to 0.7 and $\lambda_c$ ranging from 0.5 to 0.8 are more suitable hyperparameter values.

## 5 CONCLUSION

In this paper, we propose a novel approach InfoIGL to extract invariant representation for graph OOD generalization from the perspective of information theory. We view the mutual information between graphs as invariance and decompose the process of extracting invariant representation into two tasks: reducing redundant information and maximizing mutual information. The two tasks work together to ensure that the useful invariant features are fully isolated. Specifically, we utilize an attention mechanism to reduce redundancy. Then we jointly optimize the semantic-instance and instance-instance contrastive learning to maximize the mutual information, while also introducing instance constraint and hard negative mining to prevent model collapse. Theoretical analysis together with extensive experiments demonstrate the superiority of InfoIGL, highlighting its potential for real-world applications.

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

## A   RELATED WORK

**Invariant learning for graph OOD.** Invariant learning (Zhao et al., 2019; Rosenfeld et al., 2020; Yang et al., 2020; Wang & Veitch, 2022; Chen et al., 2022) has gained increasing attention as a powerful tool for developing robust representations capable of withstanding distribution shifts. In the context of graph data, invariant learning has been extensively explored to improve generalization performance in OOD scenarios. Recent works have demonstrated the effectiveness of diverse approaches in identifying invariant graph features, including optimization methods (Arjovsky et al., 2019; Krueger et al., 2021; Sagawa et al., 2020; Kong et al., 2022), causal learning (Wu et al., 2022; Sui et al., 2022; Liu et al., 2022a; Fan et al., 2022), stable learning (Fan et al., 2021; Li et al., 2021), and data manipulation (Miao et al., 2022; Rong et al., 2019; Wang et al., 2021; Han et al., 2022).

**Contrastive learning and information theory.** Contrastive learning (He et al., 2020; Chen et al., 2020a;b; Khosla et al., 2020) has achieved success in aligning representations by pulling together positive pairs and pushing apart negative pairs, which is tightly related to information theory, particularly mutual information. The contrastive loss can maximize the mutual information between positive pairs from the same class, promoting the intra-class compactness and inter-class discrimination (Wang et al., 2022). Some approaches (Huang et al., 2021; Zhang et al., 2022b; Yao et al., 2022) have attempted to leverage contrastive learning in domain generalization tasks, demonstrating the potential of contrastive learning for handling OOD issues.

## B   THEORETICAL PROOFS AND DISCUSSIONS

### B.1   PROOFS.

**Proof for Theorem 3.** For a random variable $X$, its entropy function is defined as $H(X) = -\sum_i p(x_i)log_2 p(x_i)$, where $p(x_i)$ denotes the probability of $X$ taking on the value $x_i$. Given a probability distribution $p(x_1), p(x_2), ..., p(x_n), \sum_{i=1}^{n} p(x_i) = 1$, we take the derivative of the entropy function with respect to the variable $x_i$:

$$p(x_j) = 1 - \sum_{i=1}^{n-1} p(x_i) \tag{14}$$

$$\frac{\partial H}{\partial p(x_i)} = -[1 + log_2(p(x_i) - 1 - log_2(p(x_j))] = -log(\frac{p(x_i)}{1 - p(x_i) - \sum_k^{k \neq i,j} p(x_k)}) \tag{15}$$

$$\frac{\partial^2 H}{\partial p(x_i)^2} = -\frac{1}{p(x_i)} - \frac{1}{1 - p(x_i) - \sum_k^{k \neq i,j} p(x_k)} < 0 \tag{16}$$

Equationequation 15 is the first derivative of entropy, and equation equation 16 is the second derivative. Since the second derivative is negative and $\lim_{p(x_i) \to 0} \frac{\partial H}{\partial p(x_i)} \to +\infty$, $\lim_{p(x_i) \to 1} \frac{\partial H}{\partial p(x_i)} \to -\infty$, we conclude that the entropy function is a strictly convex function. To obtain the maximum point of the convex function, we introduce Lagrange multipliers $\lambda$ and construct the Lagrangian function as follows:

$$L(p(x_1), p(x_2), ..., p(x_n), \lambda) = -\sum_{i=1}^{n} p(x_i)log_2 p(x_i) + \lambda(\sum_{i=1}^{n} p(x_i) - 1) \tag{17}$$

Taking the first partial derivative of the Lagrangian function and setting it to zero yield:

$$\frac{\partial L}{\partial p(x_i)} = \frac{\partial}{\partial p(x_i)}[-\sum_{i=1}^{n} p(x_i)log_2 p(x_i) + \lambda(\sum_{i=1}^{n} p(x_i) - 1)] \tag{18}$$

$$\Rightarrow \lambda = log_2 p(x_i) + 1, p(x_i) = 2^{\lambda - 1} \tag{19}$$

$$p(x_1) = p(x_2) = ... = p(x_n) = \frac{1}{n} \tag{20}$$

This means that given probability distribution, the entropy achieves its maximum when the distribution is uniform distribution. Conversely, if significant differences exist among the value $p(x_i)$, the entropy will be lower.

The convexity of the entropy function is an important property in information theory, as it gives us insight into how to optimize $H(X)$ based on its probability distribution. Then if we assign weights to change the probability distribution of graph features, the entropy of the graph representation $H(\Phi(G))$ can be optimized to a smaller value.

**Proof for Proposition 1.** Given variables $X_1, X_2$ and noise $S_1, S_2$, $X$ is independent of $S$, we define the mutual information between the composition of $X$ and $S$ as $I(X_1, S_1; X_2, S_2)$. According to the chain rule for information, the mutual information $I(X_1, S_1; X_2, S_2)$ can be transformed as follows:

$$I(X_1, S_1; X_2, S_2) = I(S_1; X_2, S_2|X_1) + I(X_1; X_2, S_2|S_1) \tag{21}$$

Since $X$ is independent of $S$, the mutual information between $X$ and $S$ equals zero, that means: $I(X_1; S_2) = I(X_2; S_1) = I(X_1; S_1) = I(X_2; S_2) = 0$. So $I(S_1; X_2, S_2|X_1) = I(S_1; S_2|X_1)$, $I(X_1; X_2, S_2|S_1) = I(X_1; X_2|S_1)$, then we can simplify $I(X_1, S_1; X_2, S_2)$ as follows:

$$\begin{aligned} I(X_1, S_1; X_2, S_2) &= I(S_1; S_2|X_1) + I(X_1; X_2|S_1) \\ &= I(S_1; S_2) + I(X_1; X_2) \geq I(X_1; X_2) \end{aligned} \tag{22}$$

Since $I(X_1, S_1; X_2, S_2) = I(X_1, X_2) + (S_1; S_2)$, we can conclude that directly calculating the mutual information between variables with noise $I(X_1, S_1; X_2, S_2)$ may capture spurious correlations $I(S_1; S_2)$.

## B.2 DISCUSSIONS.

**Explanation for Proposition 2.** Under L2 normalization, the Euclidean distance is equivalent to the negative of cosine similarity:

$$x^\top y = 1 - \frac{1}{2}||x - y||_2^2 \tag{23}$$

We define $\{\mathbf{x}_j\}_{j=1}^K$ as the negative samples for instance $\mathbf{x}_i$. Then the contrastive loss for negative samples can be defined as:

$$\begin{aligned} \frac{1}{N}\sum_{i=1}^N -\log\frac{1}{\sum_j \exp(\mathbf{x}_i^\top \mathbf{x}_j/\tau)} &= \frac{1}{N}\sum_{i=1}^N \log e^{1/\tau}(\sum_{j=1}^K \exp(-||\mathbf{x}_i - \mathbf{x}_j||_2^2/2\tau) \\ &= \frac{1}{N}\sum_{i=1}^N \log(\sum_{j=1}^K \exp(-||\mathbf{x}_i - \mathbf{x}_j||_2^2/2\tau) + \alpha_0 \end{aligned} \tag{24}$$

The loss for negative samples aims to encourage uniformity (Wang & Isola, 2020) and will encourage the distribution of samples close to uniform distribution (Ma et al., 2023). According to Theorem 3, the uniformity of probability distribution will cause entropy to increase, which violates our task I. Thus we apply contrastive learning in another space $\mathbf{z}_G$ that is mapped by a projection head (*i.e.,* $\mathbf{z}_G = \sigma(\text{MLP}(\mathbf{h}_G))$). Due to the non-linear mapping property of MLP, the space $\mathbf{z}_G$ and the space $\mathbf{h}_G$ are in the different unit balls. Since uniformity is encouraged in the output of the projection head (*i.e.,* $\mathbf{z}_G$), the problem of entropy increase in original representation $\mathbf{h}_G$ can be alleviated.

**Explanation for Proposition 3.** For class $c$, given the instance embeddings $\{\mathbf{z}_{G_i}\}_{i=i}^{N_c}$ and the corresponding semantic $\mathbf{w}_c = \sum_{i=1}^{N_c}(m_i \cdot \mathbf{z}_{G_i})$. Based on the properties of multivariate mutual information, the mutual information $I(\sum_{i=1}^{N_c}(m_i \cdot \mathbf{z}_{G_i}); \mathbf{z}_{G_1}; ...; \mathbf{z}_{G_{N_c}})$ can be transformed as follows:

$$I(\sum_{i=1}^{N_c}(m_i \cdot \mathbf{z}_{G_i}); \mathbf{z}_{G_1}; ...; \mathbf{z}_{G_{N_c}}) = I(\mathbf{z}_{G_1}; ...; \mathbf{z}_{G_{N_c}}) - I(\mathbf{z}_{G_1}; ...; \mathbf{z}_{G_{N_c}}|\sum_{i=1}^{N_c}(m_i \cdot \mathbf{z}_{G_i})) \tag{25}$$

Since $\sum_{i=1}^{N_c}(m_i \cdot \mathbf{z}_{G_i})$ contains all the variables $\mathbf{z}_{G_i}$ and provides some knowledge about their joint distribution, so the conditional mutual information $I(\mathbf{z}_{G_1}; ...; \mathbf{z}_{G_{N_c}}|\sum_{i=1}^{N_c}(m_i \cdot \mathbf{z}_{G_i}))$ is non-negative. Then we can conclude:

$$I(\sum_{i=1}^{N_c}(m_i \cdot \mathbf{z}_{G_i}); \mathbf{z}_{G_1}; \mathbf{z}_{G_2}; ...; \mathbf{z}_{G_{N_c}}) \leq I(\mathbf{z}_{G_1}; \mathbf{z}_{G_2}; ...; \mathbf{z}_{G_N}) \tag{26}$$

Therefore, the mutual information between the semantic $\mathbf{w}_c = \sum_{i=1}^{N_c}(m_i \cdot \mathbf{z}_{G_i})$ and all graph embedding $\mathbf{z}_{G_i}$ is always less than or equal to the mutual information between $\mathbf{z}_{G_i}$: $I(\mathbf{w}_c; \mathbf{z}_{G_1}; \mathbf{z}_{G_2}; ...; \mathbf{z}_{G_{N_c}}) \leq I(\mathbf{z}_{G_1}; \mathbf{z}_{G_2}; ...; \mathbf{z}_{G_{N_c}})$.

## C TRAINING PROCEDURE

### C.1 THE TRAINING ALGORITHM

The overall training procedure of the proposed InfoIGL is summarized in Algorithm 1.

---

**Algorithm 1:** InfoIGL's learning algorithm.

---

**Input:** Training graph dataset $\mathcal{D}^{tr}$, hyperparameters $\lambda_s, \lambda_i, \lambda_c$, number of epochs $e$, batch size $b$.
**Output:** graph encoder $\Phi(\cdot)$ (GNN encoder and Attention mechanism), classifier $\theta(\cdot)$.

1: Initialize graph encoder $\Phi(\cdot)$, Projection head $f_{\theta_p}(\cdot)$, classifier$\theta(\cdot)$.
2: **for** epoch in $1, 2, ...e$ **do**
3:     Sample data batches $\mathcal{B} = \mathcal{D}^1, \mathcal{D}^2, ..., \mathcal{D}^K$ with batch size $b$ from $\mathcal{D}^{tr}$.
4:     **for** $D^i \leftarrow \{(G_k, \mathbf{y}_k)\}_{k=1}^b \subset \mathcal{B}, i \in 1...K$ **do**
5:         # Reduce redundant information
6:         Calculate $\mathbf{h}_{G_k} \leftarrow \Phi(G_k)$ .
7:         # Maximize mutual information
8:         Calculate $\mathbf{z}_{G_k} = f_{\theta_p}(\mathbf{h}_{G_k})$.                   # Projection head
9:         Aggregate $\mathbf{z}_{G_k}$ to $\mathbf{w}_c$.                   # Get category semantics
10:       Calculate the semantic-wise contrastive loss $\mathcal{L}_{\text{sem}}$.
11:       Calculate $\mathbf{z}'_{G_k} = \lambda_c \mathbf{z}_{G_k} + (1 - \lambda_c)\mathbf{w}_c$.       # Constrain instance embedding
12:       Obtain hard negative samples $\mathbf{z}'_{hard}$.             #hard negative mining
13:       Calculate the instance-wise contrastive loss $\mathcal{L}_{\text{ins}}$.
14:       # Transfer to downstream tasks
15:       Calculate the prediction loss $\mathcal{L}_{\text{pred}}$.
16:       $\mathcal{L} = \mathcal{L}_{\text{pred}} + \lambda_s \mathcal{L}_{\text{sem}} + \lambda_i \mathcal{L}_{\text{ins}}$.                 # Total loss
17:     **end for**
18: **end for**
19: Update all the trainable parameters to minimize $\mathcal{L}$

---

## D EXPERIMENTAL DETAILS

### D.1 DATASET DESCRIPTION

We adopt one synthetic dataset (Motif) and three real-world datasets (HIV, Molbbbp, CMNIST) to evaluate our model. Here we provide the introduction and statistical details for datasets.

- **Motif (Gui et al., 2022)** is a synthetic dataset motivated by Spurious-Motif (Ying et al., 2019), graphs of which are generated by connecting a base graph (wheel, tree, ladder, star, and path) and a motif (house, cycle, and crane). The motifs are invariant for prediction while the base graphs may cause distribution shifts. Here, we use the concept shift of "base" and the covariate shift of "size" to create testing datasets.

- **HIV (Gui et al., 2022)** and **Molbbbp (Hu et al., 2020)** are small-scale molecular datasets adapted from MoleculeNet (Wu et al., 2018) in the real world, where atoms serve as nodes and chemical bonds serve as edges. The data splits of the two datasets are designed based on two domain selections: "scaffold" and "size". "Scaffold" selection is based on the Bemis-Murcko scaffold, which represents the two-dimensional structural foundation of a molecule, while "size" selection involves the number of nodes in a molecular graph - an inherent structural feature of a graph. As neither feature should determine the label, both can contribute to significant distribution shifts. We select OOD testing datasets based on the concept shift of "size" and "scaffold".

- **CMNIST (Gui et al., 2022)** is a real-world dataset created by applying superpixel techniques to handwritten digits. The dataset is divided into two parts: covariate shift split and concept shift split. In the covariate shift split, digits are colored with seven different colors, with the first five colors, the sixth color, and the seventh color assigned to the training, validation, and test sets, respectively. In the concept shift split, digits are colored with 10 different colors, with each color being highly correlated with one digit label in the training set. However, colors have weak or no correlations

with labels in the validation and test sets. Here we use the covariate shift of "color" as the OOD testing dataset.

Statistics of these datasets are presented in Table 4

Table 4: Statistics of multiple datasets.

| Dataset | #Train | #Val | #Test | #Classes | #Metrics |
|---|---|---|---|---|---|
| Motif-size | 18000 | 3000 | 3000 | 3 | ACC |
| Motif-base | 12600 | 6000 | 6000 | 3 | ACC |
| HIV-size | 14454 | 9956 | 10525 | 2 | ROC-AUC |
| HIV-scaffold | 15209 | 9365 | 10037 | 2 | ROC-AUC |
| Molbbbp-size | 1631 | 204 | 204 | 2 | ROC-AUC |
| Molbbbp-scaffold | 1631 | 204 | 204 | 2 | ROC-AUC |
| CMNIST-color | 42000 | 14000 | 14000 | 10 | ACC |

## D.2 COMPARED BASELINES

We compare our method against several state-of-the-art methods that are designed for graph OOD generalization. These baselines can be classified into four categories:

- **Optimization methods** aim to design optimization objectives to enhance the robustness of the model across different environments, including ERM, IRM (Arjovsky et al., 2019), VREx (Krueger et al., 2021), GroupDRO (Sagawa et al., 2020), FLAG (Kong et al., 2022). Empirical Risk Minimization(ERM) is a principle used in machine learning for selecting models. It involves selecting the model that minimizes the empirical risk, which is the average loss over the training data. IRM (Arjovsky et al., 2019) aims to find data representations that have consistent performance across all environments by discouraging feature distributions that lead to different optimal linear classifiers for each environment. VREx (Krueger et al., 2021) focuses on both covariate robustness and invariant prediction, and reduces risk variance in test environments by minimizing the variances of risks in training environments. GroupDRO (Sagawa et al., 2020) uses fair optimization to address the issue of insufficient training for minority distributions. Additionally, it employs risk interpolation, which involves minimizing the loss in the worst training environment explicitly. FLAG (Kong et al., 2022) is achieved by applying gradient-based adversarial perturbations to the node features, which helps the model maintain robustness to fluctuations in input data and improve prediction performance.

- **Causal learning** utilizes causal theory to extract causal features that play a key role in model prediction and ignore non-causal features, including DIR (Wu et al., 2022), CAL (Sui et al., 2022), GREA (Liu et al., 2022a), CIGA (Chen et al.), Disc (Fan et al., 2022); DIR (Wu et al., 2022) chooses a subset of graph representations as causal rationales, and performs interventional data augmentation to generate multiple distributions. CAL (Sui et al., 2022), GREA (Liu et al., 2022a) and DisC (Fan et al., 2022) are designed based on the Structural Causal Models (SCM) to disentangle causal subgraphs. While CIGA (Chen et al.) maximizes the mutual information between causal graphs and labels to extract causal-invariant subgraphs.

- **Stable learning** is committed to independently extracting stable features across different environments by means of sample reweighting, such as StableGNN (Fan et al., 2021), OOD-GNN (Li et al., 2021); StableGNN (Fan et al., 2021) proposes a novel causal variable-distinguishing regularizer which can eliminate correlations between each pair of high-level variables through a set of sample weights while OOD-GNN (Li et al., 2021) can eliminate statistical dependencies between variant and invariant representations by using randomized Fourier features and sample reweighting in a nonlinear graph representation.

- **Data manipulation** is dedicated to generating diverse augmented data to increase the diversity of data distribution, such as GSAT (Miao et al., 2022), DropEdge (Rong et al., 2019),

M-Mixup (Wang et al., 2021), G-Mixup (Han et al., 2022). GSAT (Miao et al., 2022) injects randomness to prevent unrelated information from interfering with the labels, and utilizes the reduction in randomness to select subgraphs relevant to the labels. DropEdge (Rong et al., 2019), M-Mixup (Wang et al., 2021) and G-Mixup (Han et al., 2022) apply various transformations to augment the graph data which is beneficial to realize graph generalization.

For ERM, IRM (Arjovsky et al., 2019), VREx (Krueger et al., 2021), GroupDRO (Sagawa et al., 2020), DIR (Wu et al., 2022), and M-Mixup (Wang et al., 2021), we obtain their performance on diverse datasets following the prior research (Gui et al., 2022). While for other baseline implementations, we use the codes released by the authors and evaluate their performance on a variety of datasets.

Additionally, **classic graph contrastive learning methods** which incorporate information theory, including CNC (Zhang et al., 2022a), GMI (Peng et al., 2020), Infograph (Sun et al., 2019), GraphCL (Hafidi et al.), are discussed as benchmarks. CNC (Zhang et al., 2022a) utilized ERM to identify samples with different spurious features and then leveraged contrastive learning to learn similar representations for samples of the same class. GMI (Peng et al., 2020) directly derived MI by comparing the input (i.e., the sub-graph consisting of the input neighborhood) and the output (i.e., the hidden representation of each node) of the encoder. Infograph (Sun et al., 2019) enhanced the alignment of graph representations by maximizing the mutual information between representations learned by InfoGraph and that acquired by existing supervised methods. GraphCL (Hafidi et al.) learned node embeddings by maximizing the similarity between representations of two randomly perturbed versions of the same node.

### D.3 DETAILS OF HYPERPARAMETERS FOR INFOIGL

Our code is implemented based on PyTorch Geometric. For all the experiments, we use the Adam optimizer, where the initial learning rate and the minimum learning rate are searched within $\{0.01, 0.001, 0.0001\}$ and $\{0.001, 0.00001, 0.000001\}$, respectively. We select embedding dimensions from $\{32, 64, 128, 300\}$ and choose batch sizes from $\{64, 128, 256, 512, 1024\}$. The dropout ratio is searched within $\{0.1, 0.3, 0.5\}$ while $\lambda_c, \lambda_s, \lambda_i$ are searched within $\{0.1, 0.2, 0.3, 0.4, 0.5, 0.6, 0.7, 0.8, 0.9\}$. We adopt grid search to tune the hyperparameters and list the details of hyperparameters for InfoIGL in Table 5.

Table 5: The hyperparameters for InfoIGL on different datasets

| hyperparameters | Motif | | HIV | | Molbbbp | | CMNIST |
|---|---|---|---|---|---|---|---|
| | size | base | size | scaffold | size | scaffold | color |
| encoder layers | 3 | 3 | 3 | 3 | 2 | 2 | 5 |
| embedding dim | 64 | 128 | 300 | 128 | 128 | 300 | 32 |
| max epoch | 200 | 200 | 100 | 200 | 100 | 100 | 150 |
| pretrain | 40 | 40 | 80 | 40 | 20 | 80 | 60 |
| batch size | 1024 | 128 | 256 | 1024 | 64 | 1024 | 256 |
| ini-lr | 0.001 | 0.001 | 0.001 | 0.01 | 0.01 | 0.0001 | 0.001 |
| min-lr | 1e-3 | 1e-6 | 1e-6 | 1e-6 | 1e-6 | 1e-6 | 1e-3 |
| weight decay | 0 | 1e-1 | 1e-2 | 1e-5 | 1e-5 | 0 | 0 |
| drop ratio | 0.5 | 0.5 | 0.3 | 0.3 | 0.3 | 0.3 | 0.5 |
| $\lambda_c$ | 0.7 | 0.7 | 0.7 | 0.7 | 0.2 | 0.7 | 0.7 |
| $\lambda_s$ | 0.8 | 0.5 | 0.5 | 0.5 | 0.2 | 0.2 | 0.5 |
| $\lambda_i$ | 0.2 | 0.5 | 0.5 | 0.1 | 0.2 | 0.2 | 0.1 |

### D.4 RESULTS OF SENSITIVITY ANALYSIS

We conduct experiments to evaluate how sensitive is InfoIGL to the choice of graph neural network architectures (GCN, GIN, and GAT). The results are listed in Table 6. As shown in Table 6, InfoIGN-GCN, InfoIGL-GIN, and InfoIGL-GAT are competent on Motif and CMNIST datasets while they far

surpass the baseline ERM. The results demonstrate the effectiveness of our method, irrespective of the choice of GNN backbones.

Table 6: Results of experiments with different backbones.

| methods | Motif | | CMNIST |
|---|---|---|---|
| | size | base | color |
| ERM | $70.75_{\pm0.56}$ | $81.44_{\pm0.45}$ | $28.60_{\pm1.87}$ |
| InfoIGL-GCN | $86.53_{\pm2.15}$ | $91.56_{\pm0.91}$ | $38.30_{\pm0.76}$ |
| InfoIGL-GIN | $85.53_{\pm2.37}$ | $92.51_{\pm0.16}$ | $38.93_{\pm1.11}$ |
| InfoIGL-GAT | $84.66_{\pm1.23}$ | $90.32_{\pm1.45}$ | $37.51_{\pm2.07}$ |

Additionally, we perform experiments to access the sensitivity of InfoIGL to its hyperparameters (*i.e.,* $\lambda_c, \lambda_s, \lambda_i$). In the main body of the paper, we only presented a line graph for sensitivity analysis, here we list the detailed results of sensitivity analysis on $\lambda_c, \lambda_s, \lambda_i$ in Table 7, Table 8, Table 9, respectively.

Each column of the table represents the experimental results of the model with varying ratios on different datasets. From the data in each column of the table, it can be observed that the model results vary slightly across different ratios, indicating that the model is not sensitive to these hyperparameters.

Table 7: Performance of InfoIGL with different constraint ratio $\lambda_c$.

| ratio | Motif | | HIV | | Molbbbp | | CMNIST |
|---|---|---|---|---|---|---|---|
| | size | base | size | base | scaffold | scaffold | color |
| 0.1 | $82.53_{\pm2.75}$ | $90.46_{\pm1.13}$ | $87.73_{\pm2.40}$ | $70.89_{\pm2.12}$ | $83.07_{\pm1.24}$ | $79.44_{\pm3.24}$ | $38.34_{\pm1.78}$ |
| 0.2 | $84.13_{\pm1.79}$ | $89.86_{\pm0.70}$ | $89.42_{\pm2.60}$ | $70.47_{\pm2.12}$ | $83.39_{\pm2.76}$ | $80.45_{\pm3.22}$ | $37.13_{\pm3.37}$ |
| 0.3 | $84.18_{\pm2.05}$ | $89.66_{\pm1.10}$ | $89.55_{\pm2.98}$ | $71.16_{\pm1.41}$ | $81.38_{\pm1.65}$ | $82.34_{\pm1.04}$ | $37.95_{\pm2.78}$ |
| 0.4 | $84.23_{\pm1.65}$ | $89.84_{\pm0.76}$ | $90.84_{\pm2.31}$ | $68.97_{\pm3.40}$ | $82.93_{\pm1.71}$ | $72.56_{\pm0.51}$ | $36.49_{\pm2.22}$ |
| 0.5 | $83.62_{\pm2.38}$ | $89.21_{\pm1.68}$ | $91.18_{\pm1.38}$ | $69.81_{\pm2.50}$ | $81.18_{\pm2.86}$ | $72.69_{\pm2.96}$ | $39.05_{\pm2.05}$ |
| 0.6 | $83.48_{\pm1.80}$ | $90.67_{\pm0.69}$ | $91.36_{\pm1.50}$ | $69.64_{\pm2.64}$ | $82.03_{\pm1.37}$ | $73.70_{\pm2.02}$ | $38.92_{\pm1.52}$ |
| 0.7 | $83.65_{\pm1.05}$ | $90.39_{\pm0.83}$ | $92.55_{\pm1.02}$ | $69.60_{\pm2.27}$ | $79.98_{\pm3.01}$ | $77.36_{\pm2.02}$ | $38.51_{\pm3.22}$ |
| 0.8 | $85.05_{\pm3.30}$ | $90.86_{\pm0.87}$ | $92.75_{\pm0.84}$ | $71.02_{\pm1.81}$ | $80.07_{\pm2.91}$ | $80.14_{\pm1.24}$ | $38.01_{\pm1.70}$ |
| 0.9 | $82.67_{\pm2.44}$ | $90.86_{\pm0.77}$ | $93.23_{\pm1.01}$ | $71.04_{\pm2.54}$ | $80.12_{\pm3.25}$ | $75.71_{\pm1.27}$ | $33.02_{\pm2.85}$ |

Table 8: Performance of InfoIGLL with different semantic loss ratio $\lambda_s$

| ratio | Motif | | HIV | | Molbbbp | | CMNIST |
|---|---|---|---|---|---|---|---|
| | size | based | size | scaffold | size | scaffold | color |
| 0.1 | $81.39_{\pm2.84}$ | $90.28_{\pm0.57}$ | $92.24_{\pm1.26}$ | $69.61_{\pm1.16}$ | $80.56_{\pm2.51}$ | $74.89_{\pm2.30}$ | $37.05_{\pm2.15}$ |
| 0.2 | $81.17_{\pm3.40}$ | $90.83_{\pm1.00}$ | $92.09_{\pm1.07}$ | $71.16_{\pm1.77}$ | $81.43_{\pm1.98}$ | $75.38_{\pm0.89}$ | $38.14_{\pm2.43}$ |
| 0.3 | $83.85_{\pm3.10}$ | $90.57_{\pm0.81}$ | $92.14_{\pm0.65}$ | $70.72_{\pm0.81}$ | $80.81_{\pm2.41}$ | $75.38_{\pm2.45}$ | $36.64_{\pm1.87}$ |
| 0.4 | $84.64_{\pm1.91}$ | $90.19_{\pm0.73}$ | $92.25_{\pm0.89}$ | $71.30_{\pm0.96}$ | $82.05_{\pm3.13}$ | $76.15_{\pm2.56}$ | $37.90_{\pm0.64}$ |
| 0.5 | $81.93_{\pm2.98}$ | $90.36_{\pm0.36}$ | $92.44_{\pm1.16}$ | $70.60_{\pm0.92}$ | $80.45_{\pm2.12}$ | $77.26_{\pm2.11}$ | $38.12_{\pm0.49}$ |
| 0.6 | $92.71_{\pm0.95}$ | $70.37_{\pm0.82}$ | $83.80_{\pm3.73}$ | $87.87_{\pm3.28}$ | $80.61_{\pm4.34}$ | $77.01_{\pm1.54}$ | $37.30_{\pm2.82}$ |
| 0.7 | $83.17_{\pm3.54}$ | $90.80_{\pm0.42}$ | $91.90_{\pm0.51}$ | $70.42_{\pm0.54}$ | $80.76_{\pm2.64}$ | $76.62_{\pm0.60}$ | $34.72_{\pm3.41}$ |
| 0.8 | $84.06_{\pm2.44}$ | $90.61_{\pm1.15}$ | $92.21_{\pm0.81}$ | $70.54_{\pm1.84}$ | $81.38_{\pm2.11}$ | $75.12_{\pm1.64}$ | $37.55_{\pm0.46}$ |
| 0.9 | $84.69_{\pm4.12}$ | $90.47_{\pm0.27}$ | $91.96_{\pm0.73}$ | $71.74_{\pm1.67}$ | $77.89_{\pm3.99}$ | $75.44_{\pm2.20}$ | $36.50_{\pm2.77}$ |

Table 9: Performance of InfoIGL with different instance loss ratio $\lambda_i$.

| ratio | Motif | | HIV | | Molbbbp | | CMNIST |
|---|---|---|---|---|---|---|---|
| | size | base | size | scaffold | size | scaffold | color |
| 0.1 | $82.57_{\pm 0.57}$ | $90.74_{\pm 0.76}$ | $91.86_{\pm 0.69}$ | $62.03_{\pm 4.63}$ | $83.81_{\pm 1.75}$ | $71.12_{\pm 1.79}$ | $38.02_{\pm 0.61}$ |
| 0.2 | $84.89_{\pm 1.41}$ | $90.30_{\pm 0.77}$ | $92.57_{\pm 0.76}$ | $60.13_{\pm 3.68}$ | $77.24_{\pm 1.78}$ | $74.71_{\pm 2.17}$ | $34.51_{\pm 1.53}$ |
| 0.3 | $84.79_{\pm 2.08}$ | $90.61_{\pm 0.44}$ | $92.26_{\pm 1.23}$ | $56.67_{\pm 1.13}$ | $78.44_{\pm 3.11}$ | $68.15_{\pm 2.88}$ | $38.28_{\pm 1.75}$ |
| 0.4 | $85.46_{\pm 1.63}$ | $90.04_{\pm 1.58}$ | $92.41_{\pm 0.68}$ | $61.77_{\pm 2.81}$ | $77.90_{\pm 2.08}$ | $68.04_{\pm 5.93}$ | $34.61_{\pm 0.75}$ |
| 0.5 | $82.87_{\pm 2.60}$ | $90.11_{\pm 0.42}$ | $92.25_{\pm 1.21}$ | $55.93_{\pm 3.79}$ | $79.73_{\pm 3.74}$ | $62.76_{\pm 2.78}$ | $32.69_{\pm 3.57}$ |
| 0.6 | $83.64_{\pm 1.39}$ | $90.66_{\pm 1.03}$ | $92.42_{\pm 0.88}$ | $61.33_{\pm 3.45}$ | $77.42_{\pm 2.22}$ | $60.60_{\pm 4.97}$ | $36.48_{\pm 1.80}$ |
| 0.7 | $83.17_{\pm 1.30}$ | $90.82_{\pm 0.76}$ | $92.08_{\pm 1.09}$ | $57.09_{\pm 2.29}$ | $71.17_{\pm 6.02}$ | $57.02_{\pm 2.42}$ | $29.61_{\pm 5.93}$ |
| 0.8 | $83.89_{\pm 1.44}$ | $91.10_{\pm 0.77}$ | $92.38_{\pm 0.61}$ | $58.92_{\pm 2.68}$ | $76.11_{\pm 2.48}$ | $60.43_{\pm 3.95}$ | $30.75_{\pm 1.41}$ |
| 0.9 | $80.71_{\pm 1.60}$ | $91.17_{\pm 0.37}$ | $92.09_{\pm 0.45}$ | $59.31_{\pm 2.57}$ | $73.66_{\pm 4.99}$ | $63.73_{\pm 4.49}$ | $27.09_{\pm 5.08}$ |

## E  TIME AND SPACE COMPLEXITY ANALYSIS

Let $N$ denote the number of graphs, $n$ denote the average node number per graph, $l_G, l_A, l_P$ and $d_G, d_A, d_P$ denote the numbers of layers and the embedding dimensions in the GNN backbone, attention mechanism and projection head, respectively, $C$ denote the number of class and $K$ denote the number of hard negative samples per instance. The time complexity of the GNN backbone is $O(Nnl_Gd_G)$. For the attention mechanism, the time complexity is $O(Nnl_Ad_A)$. For the projection head, since it turns from node level to graph level, the time complexity is $O(Nl_Pd_P)$. For semantic-wise contrastive learning, the time complexity is $O(NC)$. For instance-wise contrastive learning, the time complexity is $O(NK)$. Therefore, the time complexity of the whole model is $O(N(nl_Gd_G + nl_Ad_A + nl_Pd_P) + C + K)$, and the order of magnitude is $O(Nn)$. Similarly, the space complexity is also approximately $O(Nn)$ which is about the same as baselines.

## F  LIMITATIONS AND POTENTIAL NEGATIVE SOCIAL IMPACTS

### F.1  LIMITATIONS

**Informal hard negative mining.** There are several techniques for generating hard negative samples in machine learning. One approach is to choose negative samples that closely resemble positive examples by sampling from a pool of negatives. These selected samples can be relabeled as "hard negatives" and included in the training process. Another method involves the use of sophisticated algorithms like online hard example mining (OHEM), which identifies challenging negative samples based on their loss values during training. However, instead of these methods, we select hard negative samples by computing the distance between the negative samples and the semantic center that corresponds to the positive sample. While this informal hard negative mining technique may conserve computational resources, it could also introduce a certain degree of error.

**Lack of testing the applicability to other tasks.** Contrastive learning is typically used to extract useful features that can be transferred to downstream tasks. Therefore, the InfoIGL framework, which employs contrastive learning techniques, has the potential to improve the performance of other downstream tasks such as node classification. The applicability of InfoIGL to other tasks is what we leave for future work.

### F.2  POTENTIAL NEGATIVE SOCIAL IMPACTS

As our framework can extract the invariant representations with information theory and realize reliable GNNs which can alleviate graph OOD problems, we are confident that the benefits of our work outweigh any potential negative impacts. However, the trustworthy GNN may cause over-reliance and be widely applied in the real world, leading to decreased human ability and consequent unemployment. Additionally, excessive trust in this technology may cause significant losses due

to potential error overlooking. Therefore, it is crucial that people use the framework prudently and increase supervision when applying it.

