# OpenReview forum: "InfoIGL: Invariant Graph Learning Driven by Information Theory"
_ICLR.cc/2024/Conference — ICLR 2024 Conference Withdrawn Submission_

### Official Review · Reviewer_cgj1 · 2023-10-31

**Soundness:** 2 fair
**Presentation:** 2 fair
**Contribution:** 3 good
**Rating:** 5
**Confidence:** 2

**Summary:**

The paper proposes a novel framework called InfoIGL for graph out-of-distribution (OOD) generalization. InfoIGL leverages information theory to extract invariant graph representations by decomposing the process into two tasks: reducing redundant information and maximizing mutual information. The authors demonstrate the effectiveness of InfoIGL through extensive experiments on synthetic and real-world datasets.

**Strengths:**

- The paper introduces a novel framework called InfoIGL for graph out-of-distribution (OOD) generalization, which leverages information theory to extract invariant graph representations. This approach combines the strengths of both information theory and graph learning, providing a unique perspective on the problem.

- The paper offers a comprehensive review of relevant literature, highlighting the importance of invariant learning and contrastive learning in the context of graph OOD generalization.


- The proposed InfoIGL framework is well-structured and provides a clear decomposition of the invariant feature extraction process into two tasks: reducing redundant information and maximizing mutual information. This decomposition allows for a more focused and efficient learning process.

- The authors provide thorough theoretical analyses and extensive experiments to demonstrate the effectiveness of InfoIGL. The experiments are conducted on diverse datasets, including synthetic and real-world graphs, showcasing the framework's potential for real-world applications.

**Weaknesses:**

- The optimization of the proposed InfoIGL may be difficult due to the significant impact of Task 1 on Task 2 and the extra constraints between the contrastive learning designs. The authors should clarify the training strategy, especially for parameter initialization and the meaning of attention scores. Additionally, the relationship between redundant information and overfitting should be discussed more clearly. The paper could benefit from more detailed implementation details to facilitate reproducibility.


- Limited discussions compared with other baselines, such as CAL[1], which also adopts an attention mechanism to reduce redundant information. The authors should provide a more detailed theoretical analysis to highlight the merits of InfoIGL.  The paper could include more ablation studies to further validate the significance of each component in the InfoIGL framework. The authors could provide more comprehensive comparisons with widely-known baselines in the field.


- Evaluations are only conducted on the graph classification task in limited scenarios. It would be beneficial to see experiments on more diverse scenarios (transductive and inductive settings) and tasks, such as node classification.


[1] Yongduo Sui, Xiang Wang, Jiancan Wu, Min Lin, Xiangnan He, and Tat-Seng Chua. Causal attention for interpretable and generalizable graph classification. In Proceedings of the 28th ACM SIGKDD Conference on Knowledge Discovery and Data Mining, pp. 1696–1705, 2022.

**Questions:**

Could the authors provide more details on the training strategy for the InfoIGL framework, including parameter initialization and the meaning of attention scores?

How does the proposed InfoIGL framework compare to other baselines, such as CAL, in terms of theoretical merits?

How does the performance of the InfoIGL framework on more diverse scenarios (transductive and inductive settings) and tasks, such as node classification?

---

### Official Review · Reviewer_HyhR · 2023-10-31

**Soundness:** 2 fair
**Presentation:** 3 good
**Contribution:** 2 fair
**Rating:** 3
**Confidence:** 4

**Summary:**

This paper proposes a framework called InfoIGL for graph out-of-distribution generalization, which leverages information theory to extract invariant graph representations. The framework decomposes the process into two tasks: reducing redundant information and maximizing mutual information. The framework uses attention mechanism, contrastive learning, instance constraint, and hard negative mining to achieve these tasks. The paper provides theoretical analysis and experimental results to demonstrate the effectiveness and superiority of InfoIGL.

**Strengths:**

1. This paper is generally well-written. Theoretical analysis and proofs are provided as necessary.
2. This paper decomposes the invariant feature extraction process into two tasks: reducing redundant information and maximizing mutual information, which is a novel and reasonable framework.
3. Extensive experiments prove the effectiveness of the proposed method.

**Weaknesses:**

1. The proposed method is simple, based on two contrastive learning to preserve the invariant features to improve the out-of-distribution generalization. However, the paper doesn't sufficiently discuss how the contrastive learning losses are derived from Theorem 2.
2. The paper could provide more discussion on how the attention mechanism can help to reduce the entropy of graph representations.
3. The L_cau in Figure 3 is not defined.
4. The technique contribution is limited. The attention mechanism and designed contrastive learning objectives are nothing new.

**Questions:**

1. In 3.2.3, what kind of mutual information is sacrificed or missing?
2. In Explanation for Proposition 2, can the authors rigorously prove that “Using a projection head as a springboard can avoid potential entropy increase in the original embedding”?
3. I am wondering why the authors did not use the GAT directly, as the proposed attention mechanism seems to have no difference with that in GAT.

---

### Official Review · Reviewer_isKn · 2023-11-01

**Soundness:** 3 good
**Presentation:** 3 good
**Contribution:** 2 fair
**Rating:** 3
**Confidence:** 4

**Summary:**

This paper proposes InfoIGL which utilizes information theory to extract invariant graph representations. The method addresses the challenge of out-of-distribution generalization in graph-based tasks by decomposing the process of extracting invariant features into two tasks: reducing redundant information and maximizing mutual information. The experiment results demonstrate that InfoIGL achieves state-of-the-art performance in graph classification tasks under out-of-distribution generalization. The paper contributes by introducing a framework that effectively identifies invariant features within complex graph data using information theory, thereby overcoming the limitations of existing methods that rely on data augmentations or causal disentanglement.

**Strengths:**

1. This paper utilizes information theory as a foundation for identifying invariant features within complex graph data. By viewing mutual information as the measure of invariance, the framework decomposes the process of extracting invariant features into two tasks: reducing redundant information and maximizing mutual information.

2. Good theoretical derivation and mathematical proof: in this paper, information theory, contrast learning, variational inference and other mathematical knowledge are used to gradually reduce the goal of identifying invariance on graphs, approximate calculation, decomposition into small steps that can be performed, and make it have theoretical assurance through rigorous mathematical proof.

3. State-of-the-Art performance: experiment results demonstrate that InfoIGL achieves state-of-the-art performance in graph classification tasks under out-of-distribution generalization. The framework is evaluated on diverse benchmark datasets, highlighting its effectiveness in addressing the OOD problem.

**Weaknesses:**

1. Although this paper converts the goal of identifying invariance on graphs into reducing redundant information and maximizing mutual understanding by means of approximate substitution and mathematical reduction, etc. However, these derivation methods are the application of existing works, mostly. Second, the degree of relaxation is kind of large, and thus the invariable learning is transformed into contrastive learning in general. There are questions whether the so-called invariance learned in this way is valid and should be further explained.

2. Weak contribution in methods: The core methods of InfoIGL are attention mechanism and graph contrastive learning, which are direct application of existing works and lack of necessary innovation. Apart from the theoretical derivation of the first half, all that is left of this paper is a graph contrastive learning framework, which means that the authors believe invariant identification can be achieved through contrastive learning. I think it needs further investigation.

3. Confusing experiment setups: In terms of data sets, although a benchmark dataset of graph OOD is adopted (GOOD), the selected data sets are kind of small and have less value in the real world (like CMNIST). In addition, the visualization experiment only demonstrated the embeddings of InfoIGL variants and its own, not compared to baselines. And there is a lack of visual experiments on the interpretability, i.e., which parts of the graph are more important to label predictions? This is also the problem proposed in this paper, but there is no relevant experiment.

4. Mutual information is viewed as measurement of invariance directly and heuristically. Though the theoretical proof seems solid and sufficient, I would like to see more clear and easy-to-follow examples.

**Questions:**

Please refer to the weakness above.

---

### Official Review · Reviewer_v4yT · 2023-11-02

**Soundness:** 2 fair
**Presentation:** 2 fair
**Contribution:** 2 fair
**Rating:** 3
**Confidence:** 4

**Summary:**

In this paper, the authors introduce InfoIGL, a novel framework that uses information theory to extract invariant graph representations. This approach leverages mutual information to capture graph invariance, exploiting rich semantic relations among different distributions. Experiments on synthetic and real-world datasets show that InfoIGL achieves state-of-the-art out-of-distribution (OOD) generalization in graph classification tasks.

**Strengths:**

S1: The problem of graph contrastive learning is significant.

S2: The authors conduct comprehensive theoretical analyses for these modules, focusing on information theory.

S3: I appreciate the authors for making their source code available.

**Weaknesses:**

W1: The paper's presentation requires further improvement. For instance, in Section 3.2.4, consider changing "is not uncommon" to "is common."

W2: The method's description is unclear, and some details raise questions:

(i) In Equation (7), the authors mention initializing semantic $w_{c}$ based on the average semantic representation for examples in class $c$." It's unclear how class information is obtained in a self-supervised contrastive learning setup.

(ii) Equation (9) includes the symbol ${\tau}'$, which lacks a specific definition in the paper.

(iii) The authors don't incorporate any graph features into InfoIGL. Does this mean that InfoIGL is applicable to other data domains like images and natural language?

W3: The paper uses limited datasets, and some common datasets are omitted. CAIL and GREA, two important baselines, use datasets such as TUDataset and ogbg datasets. It would be beneficial if the authors could conduct experiments on a wider range of datasets to validate their proposed method.

**Questions:**

Please see my previous comment.